# Characterizing eye-gaze positions of people with severe motor dysfunction: Novel scoring metrics using eye-tracking and video analysis

**Mari Okamoto[1], Ryosuke Kojima[1], Akihiko Ueda[1], Machiko Suzuki[1], Yasushi Okuno[1,2]***

**1** Graduate School of Medicine, Kyoto University, Kyoto, Japan, **2** Medical Sciences Innovation Hub Program, RIKEN Cluster for Science, Technology and Innovation Hub, Tsurumi-ku, Kanagawa, Japan

* okuno.yasushi.4c@kyoto-u.ac.jp

**Data Availability Statement:** All codes for scoring metrics are available from https://github.com/clinfo/TobiiEyeTrackerVideoAnalysis.git. The original patient information underlying the results

## Abstract

Nonverbal communication with people who have physical disabilities is difficult. Eye-tracking technologies have recently been developed and applied to help people with physical disabilities in their communication. However, the eye-gaze patterns of people with severe motor dysfunction (SMD) have not been analyzed in detail. To clarify characterization of people with SMD, we aimed to develop gaze position-based evaluation metrics and analyze detailed eye-gaze patterns of people with SMD. We developed two new scoring metrics: (1) saliency score based on three saliency maps—spectral residual (SR), fine grained (FG), and motion (Mo); and (2) the distance score, which represents to what extent people can chase an object in a video. The evaluation was performed on 102 participants, consisting of 35 subjects with profound intellectual and multiple disabilities (PIMD; SMD with IQ < 20), 19 with severe physical disabilities (SPD; SMD with IQ ≥ 20), and 48 healthy individuals. We observed that two saliency scores (SR and FG) and the distance score showed significant differences between the PIMD/SPD and healthy groups for the entire video, whereas Mo scores did not. Moreover, the distance score was analyzed separately for each scene, where scenes were categorized into three patterns—running, explanation, and hiding— according to the behavior of the moving objects. In the SPD and healthy groups, the explanation scenes accounted for the highest percentage of all scenes with the best distance score (63.6% and 61.9%, respectively), whereas in the PIMD group, the running scenes accounted for the highest percentage (54.5%). In conclusion, the new metrics were successful in quantitatively assessing the gaze responsiveness of people with SMD, which could not be assessed using a conventional metric, gaze-acquisition time. This study is expected to expand the possibilities of nonverbal communication using eye-tracking devices for people with SMD.

## Introduction

People with severe motor dysfunction (SMD) frequently require support from others, such as parents and helpers, and the maintenance of their quality of life depends on the caregivers

cannot be shared publicly because of the restriction of Ethics Committee.

**Funding:** Initials of the authors: M.S. Grant name: Grant-in-Aid for Scientific Research (B) Grant numbers: 17H04448 Funder name: Japan Society for the Promotion of Science URL: https://www.jsps.go.jp/english/e-grants/ The funders had no role in study design, data collection and analysis, decision to publish, or preparation of the manuscript.

**Competing interests:** The authors have declared that no competing interests exist.

such as parents and support staff [1]. People with profound intellectual and multiple disabilities (PIMD), the most severe group of SMD, have severe intellectual disability (IQ < 20) and various sensory disturbances. PIMD people are incapable of verbal communication [1–3]. Moreover, one-third of people with cerebral palsy, accounting for most people with SMD, have speech/language/communication disorders triggered by sensory disturbances, such as disturbances to the visual and auditory senses [4, 5]. Therefore, the majority of SMD use nonverbal means (e.g., line of sight, facial expression, phonation, and movement) to express their emotions and responses, and their caregivers are required to understand these nonverbal expressions to communicate with them [6–8]. However, nonverbal expressions are not easy to understand for caregivers; for example, previous studies have reported that communication with people with cerebral palsy is not understandable at 67% of the occasions, even by the familiarized caregivers [9].

Recent advances in digital devices for augmentative and alternative communication (AAC) have attracted attention as powerful tools for nonverbal communication, e.g., microswitches, voice output communication aids, and eye-tracking devices [10]. AAC devices are used to observe nonverbal behaviors of people with SMD [11, 12]. The utility of eye-tracking devices to detect behavior in people with intellectual or motor disabilities has also been reported [13, 14]. For example, a recent study on infants with Rett syndrome showed that they used eye-gaze information to predict the behaviors of the characters in a video [14]. However, all these previous studies on people with severe or multiple disabilities using eye-tracking devices have methodological problems, being mostly case-series studies with small number of patients or lacking comparisons with healthy controls. In addition, previous studies have evaluated only the acquisition time of gaze on the screen or the object, and have not evaluated the detailed gaze patterns using temporal and spatial information of gaze type, gaze position, and the object, which are the strengths of eye-tracking devices [15].

In this study, eye-tracking data measured in a relatively large number of subjects were used to compare people with SMD and healthy individuals. Moreover, we developed two new gaze position-based evaluation metrics to analyze detailed eye-gaze patterns in people with SMD. First is the saliency score, which measures the overlap between the eye-gaze position and an object specified on the saliency map, reflecting eye-gaze responses to a characteristic object in a video. Second is the distance score, which measures the distances per unit time between a moving object and the gaze position, reflecting gaze tracking to the moving object in a video. The use of these new metrics can help improve our understanding of eye-gaze responses in people with SMD.

## Materials and methods

### Participants

The present study was conducted from 2017 to 2018 at Akaimi Nursery School, Shitennoji Yawaragien, Biwako Gakuen Medical Welfare Facilities for the Disabled, Yonaha General Hospital, Kitakyushu Children's Rehabilitation Center, National Rehabilitation Center for Children with Disabilities, and Communication Support Course. A total of 119 participants (69 people with SMD and 50 healthy people) were included. The inclusion criteria were people who could maintain a lateral or sitting position for at least 5 min. Individuals with SMD, who were bedridden and had an IQ < 20 were defined as people with PIMD (PIMD group, n = 47), whereas those who were bedridden and had an IQ ≥ 20 were categorized as people with severe physical disabilities (SPD group, n = 22). Participants who were unable to open their eyes due to sleepiness or disliked using eye-gaze positions were excluded from the study. This study was approved by the ethics committee of Kyoto University Graduate School, Faculty of Medicine

(approval number: R1156), and each facility's ethics committee. Written informed consent was obtained from all the subjects or their parents before the initiation of the study.

## Data collection

We used the Tobii Pro® Spectrum eye-tracking device to collect data related to eye-gaze positions. This device was selected for the following reasons:

1. It is a nonwearable type, which allowed this experiment to be performed without imposing a burden on the participants.

2. It is portable, which allows for the experiment to be carried out in the living environment of people with SMD.

3. Due to erratic eye movements of the PIMD group, such as those that were vibration-induced, the high sampling frequency (600Hz) of this device enabled us to measure the eye movements accurately.

The experiments were conducted at the participants' facilities during their leisure time and without disturbances due to any noises around them. The device was set up in a vertical position, 60–65 cm away from the subject's face. The participants faced the monitor that displayed the video, and their eye movements and positions were captured using an infrared camera connected to the device. The duration of data collection was 52.2 second, the same as the duration of the video. One author (M.S.) performed the measurements to ensure the stability and reliability of the data.

## Video

This study used a video (ColoColoAnimal 2 ©MC(NI)/P CCA2abc) of a dog and a cat running and hiding, with Japanese audio (S1 video). Fig 1 shows the details of the video. The video was divided into seven different scenes of characters (dog and cat), their motions (explanation scenes [scenes 1, 4], running scenes [scenes 2, 5, and 6], and hiding scenes [scenes 3, 7]), and their speed for the running scenes (slow [4.31 cm/s; scene 2], moderate [14.39 cm/s; scene 5], and fast [24.90 cm/s; scene 6]).

## Dataset

We collected data consisting of the time stamp, the average eye-gaze position of the right and left eyes, and types of gazes (fixation—gaze; saccade—unconscious eye movement; unclassified—gaze type could not be distinguished; and not found—eye-gaze position could not be obtained). The time of the start of the video was stamped as 0. Gaze counts were called when the gaze-type transition occurred between fixation and unconscious eye movement or fixation and unclassified. Patients who had 0 gaze count in all scenes were excluded.

## Conventional metric

This study used the eye-gaze acquisition time based on previous studies on eye-tracking [16] as a conventional metric. The eye-gaze acquisition time was calculated by adding the number of time units where the eye-gaze positioned on the screen.

## Development of new scoring metrics

**Saliency scores.** We calculated saliency scores to analyze whether there were eye-gaze positions in the part that was characteristic of visual recognition of videos and images. The

| Scene | Illustration | The characters | Description | Running speed | Hiding place | Length (s) |
|-------|-------------|----------------|-------------|---------------|--------------|------------|
| Scene 1 | | Character A the cat | explanation | - | - | 8.11 |
| Scene 2 | | the cat | running | slow | - | 6.84 |
| Scene 3 | | Character A the cat | hiding | - | a tree | 10.78 |
| Scene 4 | | Character A the dog | explanation | - | - | 6.01 |
| Scene 5 | | the dog | running | moderate | - | 6.04 |
| Scene 6 | | the dog | running | fast | - | 3.44 |
| Scene 7 | | Character A the dog | hiding | - | grass | 10.91 |

**Fig 1. Video contents used in this study.** The video consisted of seven scenes. Scenes were categorized based on the character's motion: explanation, running, and hiding in a tree or grass. The characters' running speed in running scenes was divided into three paces; slow, moderate, and fast. The illustration in each scene was an exemplified frame.

primary process of calculating the saliency score is computing a saliency map using an image processing algorithm. We used spectral residual (SR) [17] and fine grained (FG) [18] for still images, and motion (Mo) [19] saliency maps for the video (Fig 2).

SR was displayed by subtracting the average logarithmic spectrum value from the value transformed from each image using the inverse Fourier transform method. FG emphasized main border lines for essential parts of the object using an integral image. Mo detected objects using background subtraction algorithms that selected useful backgrounds at different times. Using these saliency maps, the saliency score was computed as the following process:

Step 1: Made saliency maps for each frame using SR/FG/Mo.

Step 2: Normalized saliency from 0 to 1.

Step 3: Aggregated normalized saliency at the subject's eye-gaze position in the screen for each time slice.

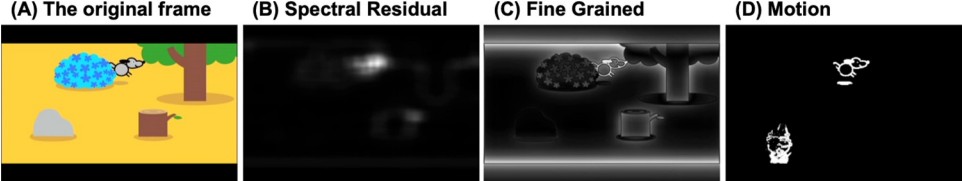

**Fig 2. Original frame and the three types of saliency maps.** (A) The original frame. (B) The Spectral Residual saliency map transformed from the original frame. (C) The Fine Grained saliency map transformed from the original frame. (D) The Motion saliency map transformed from the original frame.

Step 4: Computed a saliency score for each participant, where a saliency score was defined as the median of aggregated values for each scene or the entire video. The reference of each saliency score was calculated in eye-gaze positions at the random point on the screen at each time slice.

**Distance score.** The distance score evaluated the distance from a moving object in the video to the eye-gaze positions. This score reflects how much the participants were chasing the moving object. We have illustrated the method to detect the moving object in Fig 3. The position of the moving object was obtained as described below [20]:

Step 1: Used three consecutive frames.

Step 2: Grayscaled these frames.

Step 3: Calculated the difference between the previous frame and the current frame and between the current frame and the next frame. Then, these two differences were multiplied together.

Step 4: Performed noise reduction through dilation processing and detected connected objects.

Step 5: Calculated the center of gravity from an object. This center was defined as the moving object's position.

Step 6: Computed the distance between the moving object's position $p$ and the eye-gaze position $q$ at each time unit. This distance was measured on the screen (width: 720 pixels, height: 480 pixels). Distance score was defined as a median of Euclidian distances between $p$ and $q$ for each scene or the entire video. Note that the reference for distance score was calculated as per eye-gaze positions at the random point on the screen at each time slice.

## Statistical analysis

We calculated the eye-gaze acquisition time, saliency scores, and distance scores for the PIMD, SPD, and healthy groups. The eye-gaze acquisition time was evaluated using conventional metrics. We compared the three saliency scores and distance scores of the entire video between the three groups. We also analyzed the correlation between eye-gaze acquisition time, saliency scores, and distance scores. To characterize the relationship between the content of the video and these scores, we focused on scenes of videos described in the Materials and Methods section (Fig 1). For each scene, saliency scores were compared for the PIMD/SPD groups to detect the most useful saliency algorithm. The distance score for each scene was evaluated for participants, who had at least one gaze count for each scene, to extract the scene with the best (lowest)

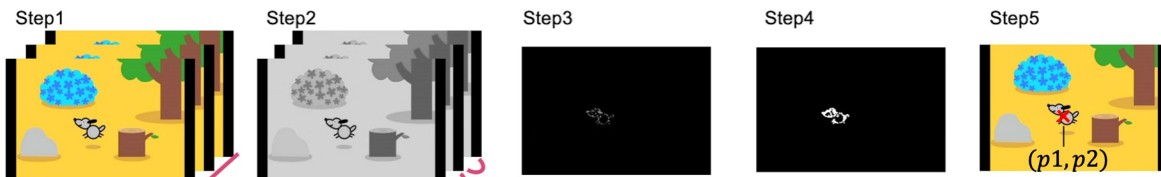

**Fig 3. How to find the motion object's positions.** Step 1: Used three consecutive frames. Step 2: Gray scaled these frames. Step 3: Calculated the difference between the previous frame and the current frame and between the current frame and the next frame. Then, these two differences were multiplied together. Step 4: Performed noise reduction by dilation processing and detected connected objects. Step 5: Calculated the center of gravity from an object. This center was defined as the moving object's position.

distance score among all scenes and running scenes (scenes 2, 5, and 6). The Kruskal–Wallis test (SciPy v1.4.1) and Dwass-Steel-Critchlow-Fligner test (scikit-posthocs 0.6.4) were used for the entire video, whereas Mann–Whitney U test (SciPy v1.4.1) was used for each scene of saliency score and Chi-square test (SciPy v1.4.1) for each scene of distance score. The significance level was set at $p < 0.05$.

## Results and discussion

### Characteristics of the subjects

Of the 119 subjects, those with no gaze count in the entire video (17 subjects, including 12 people with PIMD, three people with SPD, and two healthy people) were excluded from the analysis. Consequently, a total of 102 subjects, comprising 35 people with PIMD, 19 people with SPD, and 48 healthy individuals, were analyzed (Table 1). In the PIMD/SPD group, most participants were between 7–18 years old, whereas most participants were over 18 years old in the healthy group ($p < 0.001$). Participants who had at least one gaze count for all scenes were 11 people with PIMD, 11 people with SPD, and 42 healthy people.

### Overview of the analysis in this study

In this study, we analyzed eye-gaze patterns using eye-tracking data measured for 35 people with PIMD, 19 people with SPD, and 48 healthy people. To do this analysis, we developed two new metrics to evaluate gaze responsiveness in people with SMD. In most of the previous studies using eye tracking, only the gaze-acquisition time was used as an evaluation metric, and the relationship between the acquired gaze position and the object was not analyzed in detail. In contrast, the two new metrics we developed, saliency score and distance score, provide more detailed evaluation of the patient's gaze responsiveness compared with the gaze-acquisition time. Saliency score is the degree of coincidence between the gaze position and the object identified by the saliency map, which indexes characteristic parts such as objects in the video. Here, we used three saliency maps: spectral-residual (SR), fine-grained (FG), and motion (Mo)

**Table 1. Characteristics of study subjects and subjects with gaze count >0 for each scene.**

|  | PIMD group | SPD group | Healthy group | Total |
|---|---|---|---|---|
|  | (n = 35) | (n = 19) | (n = 48) | (n = 102) |
| **Age, n (%)** |  |  |  |  |
| $\leq 6$ | 6 (17.1) | 4 (21.1) | 15 (31.3) | 25 (24.5) |
| **7–18** | 17 (48.6) | 13 (68.4) | 8 (16.7) | 38 (37.3) |
| **18 $\leq$** | 12 (34.3) | 2 (10.5) | 25 (52.1) | 39 (38.2) |
| **Number of subjects with** |  |  |  |  |
| **gaze count >0, n (%)** |  |  |  |  |
| **All scenes** | 11 (31.4) | 11 (31.4) | 42 (87.5) | 64 (62.7) |
| **Scene 1** | 28 (80.0) | 16 (45.7) | 45 (93.8) | 89 (87.3) |
| **Scene 2** | 26 (74.3) | 17 (48.6) | 46 (95.8) | 89 (87.3) |
| **Scene 3** | 25 (71.4) | 17 (48.6) | 47 (97.9) | 89 (87.3) |
| **Scene 4** | 20 (57.1) | 18 (51.4) | 47 (97.9) | 85 (83.3) |
| **Scene 5** | 22 (62.9) | 17 (48.6) | 47 (97.9) | 86 (84.3) |
| **Scene 6** | 18 (51.4) | 16 (45.7) | 45 (93.8) | 79 (77.5) |
| **Scene 7** | 21 (60.0) | 16 (45.7) | 47 (97.9) | 84 (82.4) |

PIMD, profound intellectual and multiple disabilities; SPD, severe physical disabilities.

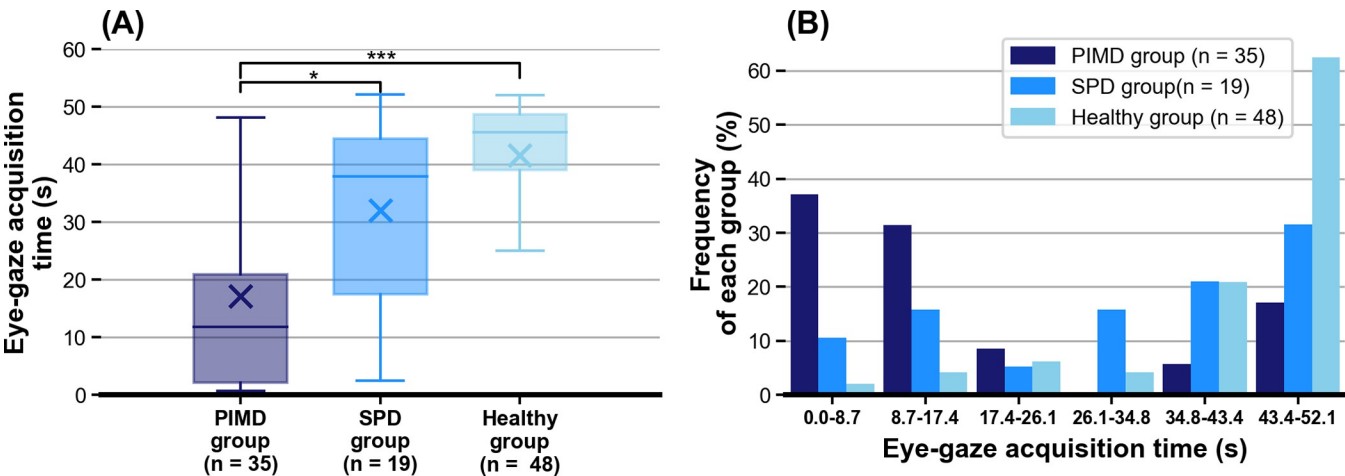

**Fig 4. Distribution for eye-gaze acquisition time.** (A) The distribution of the eye-gaze acquisition time for the entire video between the PIMD, SPD, and healthy groups is presented as Box-and-Whisker Plots. Significant statistical differences were observed between the PIMD and healthy groups and between SPD and healthy groups (p < .05), respectively. Box-and-whisker plots show median values (-), average (×), interquartile ranges, minimum and maximum values, and outliers. * p < .05, ** p < .01, *** p < .001. (B) A histogram of the distribution of the eye-gaze acquisition time for the PIMD, SPD, and healthy groups. PIMD, profound intellectual and multiple disabilities; SPD, severe physical disabilities.

saliency. Further, the distance score is the degree of coincidence between the gaze position and the moving object, which can be used to evaluate the degree of tracking of the moving object in the video. We evaluated the performance of the saliency scores and distance score in two cases, one for the entire video (S1 Video) and the other for each scene of the video (Fig 1). The results are presented in the following sections.

### Analysis of eye-gaze acquisition time as a conventional metric

To compare the conventional metrics with our new metrics, we first analyzed the eye-tracking data using gaze-acquisition time as a conventional metric. The extent to which eye gazes were successfully acquired in the entire video was evaluated for all three groups (Fig 4A). In Fig 4A, the distributions of PIMD, SPD, and healthy groups were different. Because the variances of gaze-acquisition time for PIMD and SPD groups were large, we further analyzed the frequency of the participants for each period of the eye-gaze acquisition time (Fig 4B). In Fig 4B, the healthy group showed a monotonous increasing trend, whereas the PIMD and SPD groups showed a bimodal distribution. For example, subjects who could maintain eye-gaze for more than half of the video (eye-gaze acquisition time ≥ 26.1 s) included eight patients in the PIMD group (22.9%), 13 patients in the SPD group (68.4%), and 42 participants in the healthy group (87.5%). The percentage of subjects who could maintain eye-gaze for more than half of the video in the PIMD group was lower than those of the SPD and healthy groups (p < 0.05). Alternatively, gaze-acquisition time was less than half of the video for the majority of subjects in the PIMD group, which indicates that the analyses using the eye-gaze acquisition time may underestimate the gaze performance of the PIMD group. Therefore, we used the saliency score and distance score as new evaluation metrics to assess gaze performance when gaze-acquisition time is short.

### Evaluation of the gaze responses to objects in videos using the saliency scores

In this subsection, we first show the result of evaluation using the entire video, and then explain details of scene-wise results. Fig 5 shows the distribution of saliency scores for the

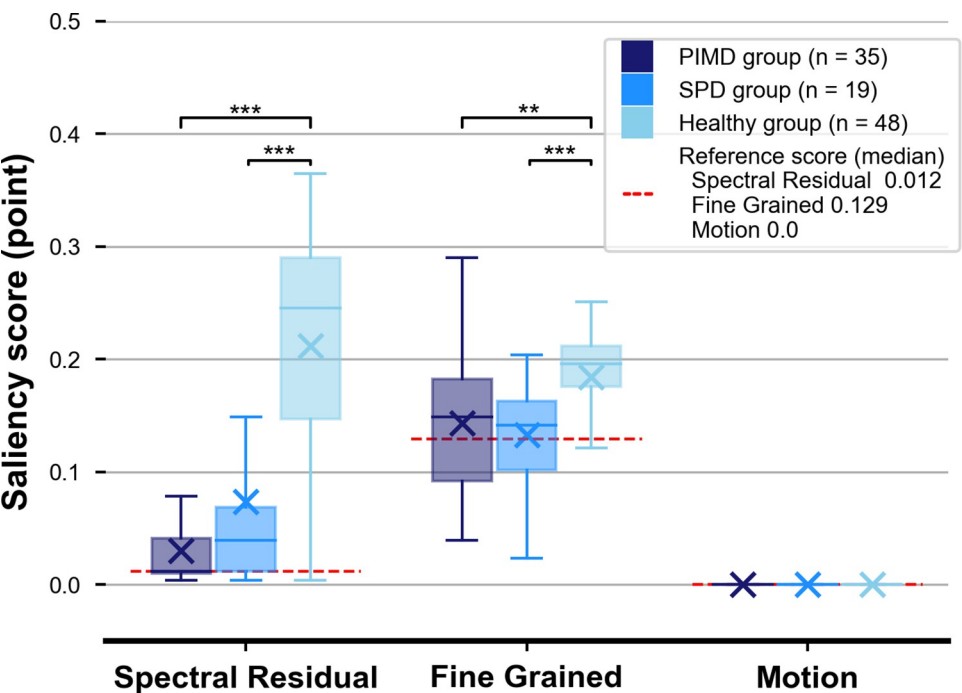

**Fig 5. Distribution of saliency scores for the entire video.** Distribution of saliency scores for the entire video among the PIMD, SPD, and healthy groups, presented as box-and-whisker plots. Significant statistical differences in spectral-residual and fine-grained scores were observed between the PIMD and healthy groups and between SPD and healthy groups (p < 0.01), respectively. Box-and-whisker plots show median values (−), average (×), interquartile ranges, minimum and maximum values, and outliers. *p < .05, ** p < .01, *** p < .001. PIMD, profound intellectual and multiple disabilities; SPD, severe physical disabilities.

entire video for the PIMD, SPD, and healthy groups. Significant statistical differences in SR and FG scores were observed between the PIMD and healthy groups and between SPD and healthy groups, respectively (p < 0.001). SR and FG scores failed to show any statistical differences between the SPD and PIMD groups. The SR and FG scores successfully stratified between PIMD/SPD groups and healthy group. For subjects with the eye-gaze acquisition time with ≥ 26.1 s, significant statistical differences were observed between PIMD/SPD groups and the healthy group (p < 0.01) regarding SR score and between SPD and healthy groups (p < 0.01) regarding FG score. In contrast, for subjects with the eye-gaze acquisition time with < 26.1 s, PIMD/SPD groups tended to have smaller SR or FG scores than those of the healthy group, although the analysis did not reach statistical significance (p = 0.08 and p = 0.32, respectively) (S1 Fig). Mo scores for the entire video were 0 for all subjects except for one healthy subject. This was due to the characteristics of the Mo scores, where zero saliency occupied most areas in each frame except limited high saliency areas on moving objects. Therefore, SR or FG scores are more useful to reflect eye-gaze responses compared with the Mo score.

Fig 6 shows the relationship between the saliency scores for the entire video and the eye-gaze acquisition time for the PIMD, SPD, and healthy groups. The healthy group had higher scores for both eye-gaze acquisition time and SR or FG scores than those of the PIMD and SPD groups. There was no correlation between eye-gaze acquisition time and SR or FG scores of each group. Note that some subjects in the PIMD and SPD groups had high saliency scores even when their eye-gaze acquisition time was low. Therefore, we suggest that the saliency score may enable to evaluate the gaze response that cannot be evaluated by conventional metrics.

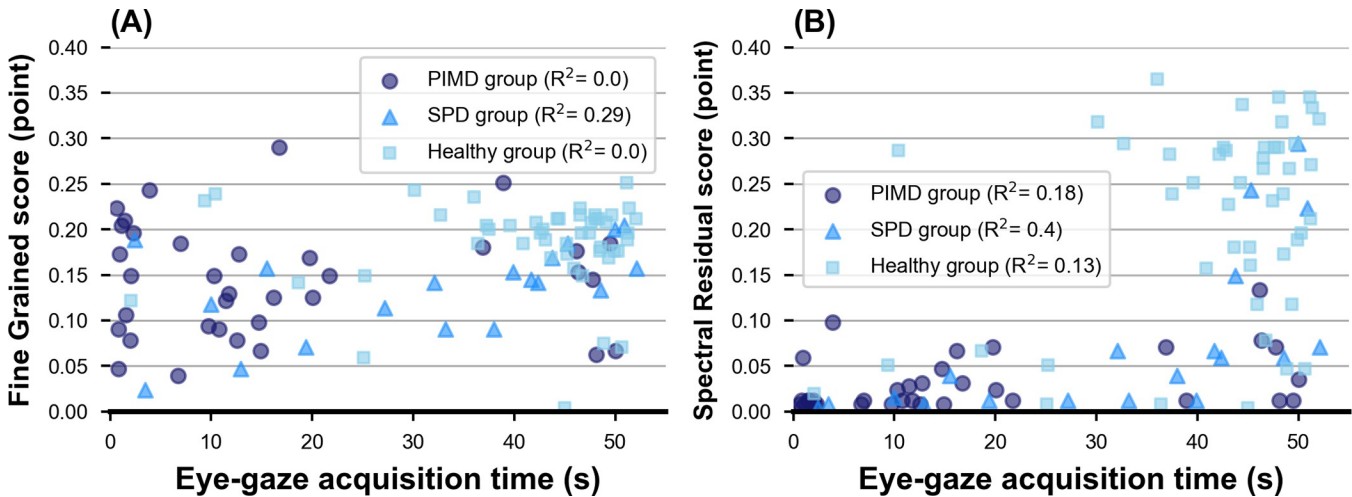

**Fig 6. Relationship between eye-gaze acquisition time and saliency scores.** (A) The relationship between eye-gaze acquisition time and fine-grained score. (B) The relationship between eye-gaze acquisition time and spectral-residual score. PIMD, profound intellectual and multiple disabilities; SPD, severe physical disabilities.

To clarify whether the SR/FG scores can be used to evaluate the eye-gaze responses of the PIMD/SPD groups to objects in the video, we analyzed the scores for each of the seven scenes (Fig 1). Fig 7 shows the distribution of SR/FG scores for each scene for the PIMD/SPD groups. The FG scores were significantly higher than the SR scores for all scenes in the PIMD group and for four scenes (scenes 2, 3, 5, and 6) for the SPD group. There was no significant difference between FG and SR scores when adjusted by the median value of the reference (0.14 ± 0.05 vs 0.08 ± 0.08 in FG and SR scores, respectively, p = 0.15). The FG saliency maps scored the area of any object on the screen, whereas the SR saliency maps scored only the location of moving objects. In the running scenes (scenes 2, 5, and 6), even if the SR score is low due to the inability to accurately track moving objects of the individuals in the PIMD and SPD groups, the FG score is useful to evaluate the responses to at least any object in the video.

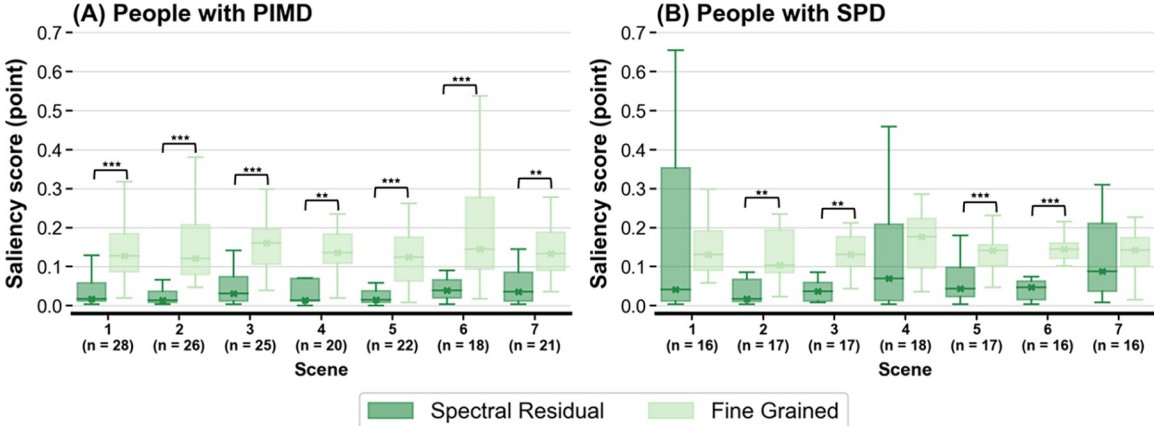

**Fig 7. Distribution of saliency scores for each scene in PIMD and SPD groups.** Distribution of spectral-residual and fine-grained scores for each scene are presented as box-and-whisker plots in (A) PIMD group and (B) SPD group. The fine-grained scores were significantly higher than the spectral-residual scores for all scenes for the PIMD group and four scenes (scenes 2, 3, 5, and 6) for the SPD group. Box-and-whisker plots show median values (−), average (×), interquartile ranges, minimum and maximum values, and outliers. *p < .05, **p < .01, *** p < .001. PIMD, profound intellectual and multiple disabilities; SPD, severe physical disabilities.

## Evaluation of the gaze responses to moving objects in videos using the distance score

This subsection demonstrates the three results related to the distance score. The former two results are the same with the previous subsection of the saliency scores: entire video and scene-wise results. In addition, a result associated with speeds of moving objects is described.

First, we show the distribution of distance scores for the entire video in the PIMD, SPD, and healthy groups in Fig 8. Significant statistical differences were found among all three groups (SPD group vs. healthy group: $p < 0.001$, PIMD group vs. healthy group: $p < 0.001$, and SPD group vs. PIMD group: $p = 0.020$). For subjects with the eye-gaze acquisition time with $\geq 26.1$ s, significant statistical differences were observed between PIMD/SPD groups and healthy group ($p < 0.01$). In contrast, for subjects with the eye-gaze acquisition time with $< 26.1$ s, PIMD/SPD groups tended to have higher distance scores compared with the healthy group, although the analysis did not reach statistical significance ($p = 0.10$) (S2 Fig). The distance score evaluates the distance between the position of the moving object and the eye-gaze position, with greater scores in the PIMD and SPD groups indicating that their eye gazes and position of the moving objects are father away.

Fig 9A shows the relationship between distance score for the entire video and the eye-gaze acquisition time for the PIMD, SPD, and healthy groups. The subjects in the healthy group had better scores both in eye-gaze acquisition time and distance scores than those of the PIMD and SPD groups. There was no correlation between eye-gaze acquisition time and distance scores for each group. Some subjects in the PIMD and SPD groups had better distance scores

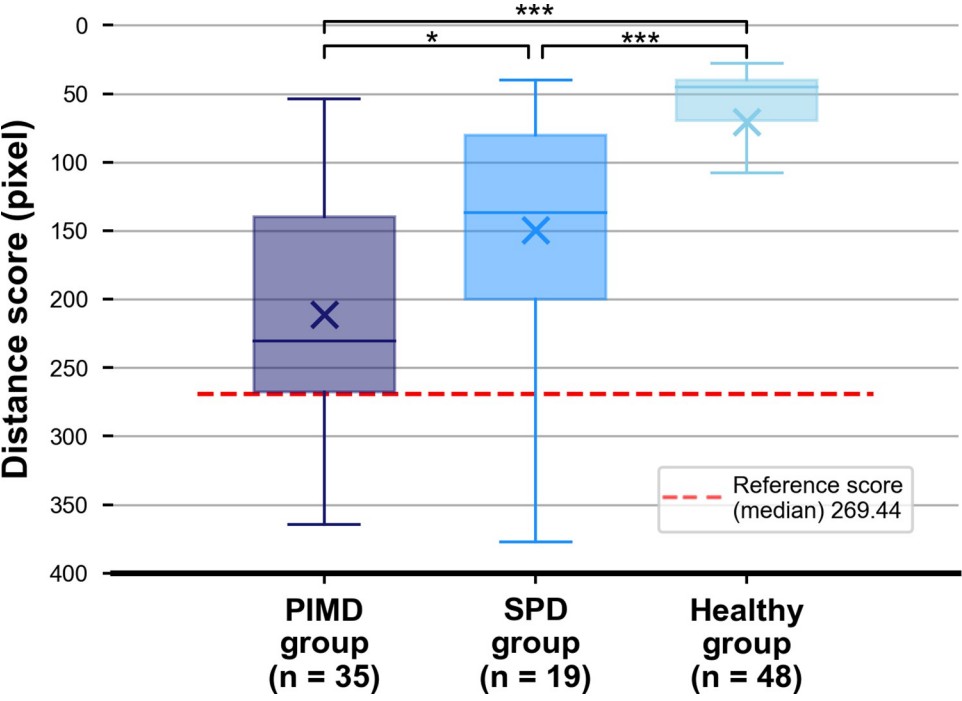

**Fig 8. Distribution of distance score for the entire video.** Distribution of distance score for the entire video for the PIMD, SPD, and healthy groups is presented as box-and-whisker plots. Significant statistical differences were observed among all three groups (SPD group vs. healthy group: $p < 0.001$, PIMD group vs. healthy group: $p < 0.001$, and SPD group vs. PIMD group: $p = 0.020$). Box-and-whisker plots show median values (−), average (×), interquartile ranges, minimum and maximum values, and outliers. *$p < .05$, ** $p < .01$, *** $p < .001$. PIMD, profound intellectual and multiple disabilities; SPD, severe physical disabilities.

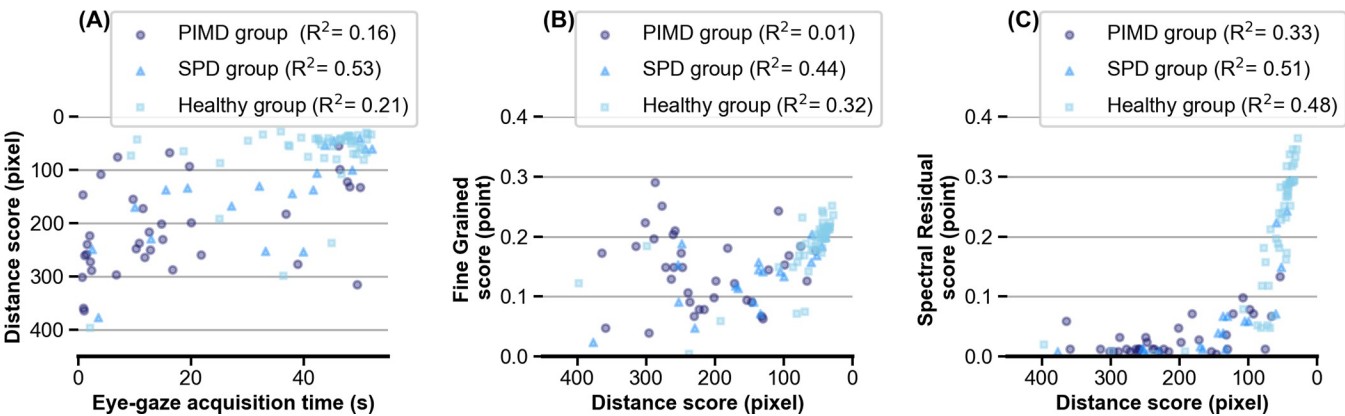

**Fig 9. Relationship between distance score and eye-gaze acquisition time or saliency scores.** (A) The relationship between eye-gaze acquisition time and distance score. (B) The relationship between distance score and fine-grained score. (C) The relationship between distance score and spectral-residual score. PIMD, profound intellectual and multiple disabilities; SPD, severe physical disabilities.

even when their eye-gaze acquisition time was low, indicating that distance scores can be used to assess subjects who cannot be evaluated by conventional metrics. Moreover, the relationship between distance score and saliency scores were analyzed for the entire video for the PIMD, SPD, and healthy groups. The correlation coefficient between FG and distance score was lower for the PIMD group ($R^2 = 0.01$) than for SPD ($R^2 = 0.44$) and healthy ($R^2 = 0.32$) groups (Fig 9B). The correlation coefficient between SR and distance score was also lower in the PIMD group ($R^2 = 0.33$) than in the SPD ($R^2 = 0.51$) and healthy groups ($R^2 = 0.48$). The distribution range of distance score, however, was wider than that of the SR score for the PIMD group, suggesting that the distance score may have higher classification performance than the SR score (Fig 9C).

To analyze the gaze tracking of moving objects in each of the seven scenes (Fig 1) in the PIMD/SPD groups, scenes where the PIMD and SPD groups had their best (lowest) distance scores were evaluated. Table 2 describes the distribution of scenes with the best distance scores in the PIMD, SPD, and healthy groups, respectively. In the PIMD group, the frequency of the best score scenes was highest for the running scenes (scenes 2, 5, and 6; 45.5%), followed by the hiding scenes (scenes 3 and 7; 18.2%) and the explanation scenes (scenes 1 and 4; 36.4%). For the SPD and healthy groups, the frequency of the best score scenes was higher for the explanation scenes (scenes 1 and 4; 63.6% vs. 64.3%) than that for the running scenes (scenes 2, 5, and 6; 36.3% vs. 16.7%) and the hiding scenes (scenes 3 and 7; 0% vs. 19.1%). People with PIMD have a lower IQ and may have more difficulty understanding language compared with SPD and healthy people. This may have led to the difference in eye-gaze patterns for the PIMD group, who were more responsive to a running object compared with to an explanation scene without movement.

**Table 2. Characteristics of study subjects' group of the best distance score by each description.**

|  | Explanation (scenes 1 and 4) | Running (scenes 2, 5, and 6) | Hiding (scenes 3 and 7) |
|---|---|---|---|
| **PIMD group, n (%)** | 4 (36.4) | 5 (45.5) | 2 (18.2) |
| **SPD group, n (%)** | 7 (63.6) | 4 (36.3) | 0 (0.0) |
| **Healthy group, n (%)** | 27 (64.3) | 7 (16.7) | 8 (19.1) |

PIMD, profound intellectual and multiple disabilities; SPD, severe physical disabilities.

**Table 3. Characteristics of subjects of the study's group with the best distance score based on running scenes.**

|  | Scene 2 (slow) | Scene 5 (moderate) | Scene 6 (fast) |
|---|---|---|---|
| **PIMD group, n (%)** | 4 (36.4) | 5 (45.5) | 2 (18.2) |
| **SPD group, n (%)** | 5 (45.5) | 6 (54.5) | 0 (0.0) |
| **Healthy group, n (%)** | 33 (78.6) | 9 (21.4) | 0 (0.0) |

PIMD, profound intellectual and multiple disabilities; SPD, severe physical disabilities.

Finally, we compared the distribution of scenes with the best distance scores among the three running scenes with different speeds (scene 2 [slow], scene 5 [moderate], and scene 6 [fast]) (Table 3). The frequency of the best score scenes was highest for scene 5 in the PIMD and SPD groups (PIMD group: 45.5%, SPD group: 54.5%), whereas that of scene 2 (78.6%) was highest for the healthy group (p < 0.01). As shown in the healthy group, a slower speed is more likely to be followed by gaze. In contrast, in the PIMD/SPD groups, moderate pace was best to obtain the best distance scores. This result indicates that appropriate speeds for good eye-gaze response are clearly different for people with PIMD/SPD compared with the healthy subjects.

To support the forementioned results, S3 Fig shows plots of the trajectory of moving objects and eye-gaze positions during scenes 2, 5, and 6 for representative subjects from the PIMD (Ba-Bc)/SPD(Ca-Cc) groups and healthy group (Da-Dc) showing the best distance scores for scenes 5 and 2, respectively. For all three groups, the gaze positions were plotted overlapping with the trajectory of the moving objects, showing that the distance score represented the ability of the eye tracking the moving objects.

## Advantages and limitations of the eye-tracker device in evaluating the gaze responses of people with SPD and PIMD

It is challenging to evaluate eye-gaze responses in people with SPD and PIMD with the same accuracy as healthy subjects. The most confounding factors affecting accuracy are as follows: 1) erratic eye movements are often observed in people with PIMD so that the gaze position shifts with instantaneous amplitude, 2) people with SPD and PIMD are usually hard to follow instructions, 3) people with SPD and PIMD have difficulty maintaining interest in particular objects and need to complete the measurements in a short time. Therefore, to analyze gaze positions in SPD and PIMD patients, eye-tracking devices with high sampling frequency and short, objective observations in an environment suitable for measurement are essential to ensure accuracy. Although eye-tracking could be difficult for conventional devices with low sampling frequency, we used Tobii Pro® Spectrum eye-tracking device to analyze eye-gaze positions in the present study. The tracking system latency of the device was 50 μs that enabled high sampling frequency faster than erratic eye movements (S1 Table).

Temporal calibration and offsets of eye-gaze positions in people with SPD and PIMD are matters, as people with SPD and PIMD have difficulty looking at the instructed points. Therefore, calibration and offset settings were performed by a healthy person instead of each subject. Intergroup comparisons were carried out to reduce measurement error due to the device. In addition, a single observer performed measurements using the same device and video to minimize the measurement time and errors on the observer side. On the other hand, the measurement conditions of the subject differed from person to person because each subject had a different clinical background and spent leisure time differently. So far, no studies have addressed the objective evaluation of the gaze characteristics of people with SPD and PIMD. Although the limitation of measurement errors, the present study has, for the first time,

succeeded in obtaining and evaluating the eye-gaze positions and patterns of people with SPD and PIMD, which showed the significance of our novel scoring metrics using eye-tracking and video analysis. The present study was designed to analyze the eye-gaze position and patterns of moving objects that attracted the subject's interests. The cognitive functions were not addressed, as a distinct position in the face was so close (e.g., the distance between the eye and mouth was about 15 pixels on a 480 x 720-pixel screen) that precise evaluation of what were stared had not been performed precisely. A future investigation may be necessary to conduct using video images focused on the specific area of interest in the objects.

### Prospects for the use of new methods for supporting communication with people with PIMD/SPD

Families and caregivers of people with SMD are required to respond appropriately to patients' reactions to improve communication and interactions [21]; however, thus far, they have not had a sensible method to evaluate patients' eye-gaze responsiveness. This study, for the first time, has quantitatively evaluated how people with PIMD look at characteristic objects or to what extent they traced moving objects. Using our proposed metrics, families and caregivers can utilize the eye-gaze pattern of people with SMD to improve communication abilities. Moreover, a previous study has reported that the gaze performance for tasks improved after 5–10 months of training with an eye tracker for people with SMD [11, 22]. If the quantitative improvement of eye-gaze responsiveness can be achieved through daily practice, it could be used as an achievable goal for better understanding of and communication with people with SMD.

### Strengths and limitations

The strengths of this study are as follows: this study included a large number of people with PIMD/SPD (54 subjects in total); experiments were conducted in the living environments of the PIMD and SPD groups to minimize the burden on the patients; and the experimental task was just to watch a video, something that people with SMD do daily [23]. Despite these strengths, the study had two limitations. First, there were variations in the clinical backgrounds, comorbidities, visual acuity, and experience with viewing videos. Second, measurement errors for each subject and device accuracy might occur because of the difference in individual measurement conditions as discussed above. Therefore, combinations of the proposed scores with subjects' background and further studies with bigger samples are warranted to develop an evaluation score tailored to individual situations in people with SMD.

## Conclusion

We developed two novel scoring metrics—saliency and distance scores—that are capable of quantitatively analyzing the eye-gaze patterns. The new metrics successfully quantified the gaze responsiveness of people with SMD, which cannot be otherwise assessed by a conventional metric. The saliency score and the distance score are useful to stratify the subjects' eye-gaze pattern, whereas the distance score further identified the scenes where people with SMD reacted. This study could lead to the expansion of the possibilities of nonverbal communication using eye-tracking devices for people with SMD.

## Supporting information

**S1 Video. A video of a dog and a cat running and hiding, with Japanese audio used in the study.**
(TIFF)

**S1 Fig. Relationship between saliency scores and eye-gaze acquisition stratified by eye-gaze acquisition time.** Distribution of saliency scores for the entire video for the PIMD, SPD, and healthy groups are presented as box-and-whisker plots, under the conditions of eye-gaze acquisition time of (A) < 26.1 s and (B) ≥ 26.1 s. Box-and-whisker plots show median values (−), average (×), interquartile ranges, minimum and maximum values, and outliers. $^*p < .05$, $^{**}p < .01$, $^{***}p < .001$. PIMD, profound intellectual and multiple disabilities; SPD, severe physical disabilities.
(TIF)

**S2 Fig. Relationship between distance scores and eye-gaze acquisition stratified by eye-gaze acquisition time.** Distribution of distance scores for the entire video for the PIMD, SPD, and healthy groups are presented as box-and-whisker plots, under the conditions of eye-gaze acquisition time of (A) < 26.1 s and (B) ≥ 26.1 s. Box-and-whisker plots show median values (−), average (×), interquartile ranges, minimum and maximum values, and outliers. $^*p < .05$, $^{**} p < .01$, $^{***}p < .001$. PIMD, profound intellectual and multiple disabilities; SPD, severe physical disabilities.
(TIF)

**S3 Fig. Plots of the locations of moving objects' positions and eye-gaze positions in scenes 2, 5, and 6 for representative subjects.** Moving objects' position (x) and eye-gaze position (o) were plotted in (a) scene 2, (b) scene 5, and (c) scene 6 for representative subjects. A color gradient represents the start (blue) and the end (red) points of the scene. (A) A moving objects' trajectory. (B) Eye-gaze positions of a person with PIMD who had the best distance score for scene 5. (C) Eye-gaze positions of a person with SPD who had the best distance score for scene 5. (D) Eye-gaze positions of a healthy subject who had the best score for scene 2.
(TIF)

**S1 Table. Specifications of Tobii Pro$^®$ Spectrum eye-tracking device.**
(DOCX)

## Author Contributions

**Conceptualization:** Machiko Suzuki, Yasushi Okuno.

**Data curation:** Mari Okamoto.

**Formal analysis:** Mari Okamoto, Ryosuke Kojima, Akihiko Ueda, Machiko Suzuki, Yasushi Okuno.

**Funding acquisition:** Machiko Suzuki.

**Investigation:** Ryosuke Kojima, Akihiko Ueda, Machiko Suzuki, Yasushi Okuno.

**Methodology:** Mari Okamoto, Ryosuke Kojima, Akihiko Ueda.

**Project administration:** Yasushi Okuno.

**Resources:** Machiko Suzuki.

**Software:** Mari Okamoto, Ryosuke Kojima.

**Supervision:** Yasushi Okuno.

**Validation:** Mari Okamoto, Ryosuke Kojima, Akihiko Ueda, Yasushi Okuno.

**Writing – original draft:** Mari Okamoto, Ryosuke Kojima, Akihiko Ueda.

**Writing – review & editing:** Mari Okamoto, Ryosuke Kojima, Akihiko Ueda, Machiko
  Suzuki, Yasushi Okuno.

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
