## [Decision Letter · Decision Letter 0]

10 Sep 2021

PONE-D-21-25819Characterizing eye-gaze positions of people with severe motor dysfunction: novel scoring metrics using eye-tracking and video analysisPLOS ONE

Dear Dr. Okuno,

Thank you for submitting your manuscript to PLOS ONE. After careful consideration, we feel that it has merit but does not fully meet PLOS ONE’s publication criteria as it currently stands. Therefore, we invite you to submit a revised version of the manuscript that addresses the points raised during the review process.

We look forward to receiving your revised manuscript.

Kind regards,

Felix Albu, Ph.D.

Academic Editor

PLOS ONE

Journal Requirements:

https://journals.plos.org/plosone/s/file?id=ba62/PLOSOne_formatting_sample_title_authors_affiliations.pd

Additional Editor Comments:

The authors should address the comments of the reviewers.

Reviewers' comments:

Reviewer's Responses to Questions

**Comments to the Author**

1. Is the manuscript technically sound, and do the data support the conclusions?

Reviewer #1: Partly

Reviewer #2: Yes

2. Has the statistical analysis been performed appropriately and rigorously? 

Reviewer #1: Yes

Reviewer #2: Yes

3. Have the authors made all data underlying the findings in their manuscript fully available?

Reviewer #1: No

Reviewer #2: Yes

4. Is the manuscript presented in an intelligible fashion and written in standard English?

Reviewer #1: Yes

Reviewer #2: Yes

5. Review Comments to the Author

Reviewer #1: Dear Authors, the article in general sounds correct, but I have found a few unaddressed issues: 1. Did you consider the system's accuracy? According to the producer's website, the highest accuracy at optimal conditions is 0.3 degrees. That is almost 4 mm at a 65 cm distance, it might be possible that in your case, the error is much higher (were "optimal conditions" assured?). How does it affect the final results? 2. There is no information about the tracking system latency. Did you perform a temporal calibration? Is the offset constant? 3. Could you elaborate on whether the moving object's position should be defined as the object's center of gravity? I can imagine the situation that the watcher is staring at an eccentric part of an object all the time, e.g., the dog's eye or nose.

Reviewer #2: The research presents a quantified approach for assessment of gaze for people with motor disabilities. The research can be extended to analyze gaze of any individual based on saliency score.

The language of presentation is clear.

The data analysis in the present work show difference of saliency score between healthy individuals and people with MD.

However the technique of video analysis (technique of calculating scoring metrics between video frames) seems cumbersome.

6. PLOS authors have the option to publish the peer review history of their article (what does this mean?). If published, this will include your full peer review and any attached files.

Reviewer #1: No

Reviewer #2: No

---

## [Author Response · Author response to Decision Letter 0]

2 Nov 2021

Reviewer #1: 

Response: 

We appreciate the time and effort the reviewer has dedicated to providing insightful feedback on ways to strengthen our paper. We have incorporated changes that reflect the detailed suggestions you have graciously provided. We hope that our edits and the responses we provide below satisfactorily address all the concerns the reviewer has noted.

"Dear Authors, the article in general sounds correct, but I have found a few unaddressed issues:"

We deeply appreciate for your positive feedback. 

"1. Did you consider the system’s accuracy? According to the producer’s website, the highest accuracy at optimal conditions is 0.3 degrees. That is almost 4 mm at a 65 cm distance, it might be possible that in your case, the error is much higher (were “optimal conditions” assured?). How does it affect the final results?"

Thank you for pointing out a crucial issue. As pointed out by the reviewer, the accuracy and measurement conditions of eye tracking are essential issues when using eye-tracker devices. In this study, a single observer performed measurements using the same device and video to minimize the measurement error on the observer side. On the other hand, the measurement conditions of the subject differed from person to person because each subject had a different clinical background and spent leisure time differently. Therefore, there may be measurement errors for each subject as well as the device accuracies. 

So far, no studies have addressed the objective evaluation of the gaze characteristics of people with SPD and PIMD. Although the limitation of measurement errors, the present study has, for the first time, succeeded in obtaining and evaluating the eye-gaze positions and patterns of people with SPD and PIMD, which showed the significance of our novel scoring metrics using eye-tracking and video analysis. We added following sentences on measurement errors to paragraph of the Strengths and limitations.

Line 456–459:

First, there were variations in the clinical backgrounds, comorbidities, visual acuity, and experience with viewing videos. Second, measurement errors for each subject and device accuracy might occur because of the difference in individual measurement conditions.

"2. There is no information about the tracking system latency."

We apologize for the insufficient description. The tracking system latency of the device is 50 µs which is less than timestamps acquisition duration (1670 µs). We have added the manuscripts in the paragraph of Method section as follows, in accordance with the reviewer’s suggestion.

Line 108–109:

The tracking system latency of the device was 50 µs which is less that sampling frequency.

"Did you perform a temporal calibration? Is the offset constant?" 

Thank you for pointing out an essential issue. It is not easy to perform accurate calibration of eye-gaze positions in people with SPD and PIMD as they are usually hard to follow the instructions. Moreover, it would be a burden on patients to continue the experiment until accurate calibration becomes possible. Therefore, in people with SPD and PIMD, calibration was performed by a healthy person instead of each subject. Although calibration can affect the accuracy and offset, as the reviewer is concerned, we believe that this does not affect the importance of the main findings of this study.

"3. Could you elaborate on whether the moving object’s position should be defined as the object’s center of gravity? I can imagine the situation that the watcher is staring at an eccentric part of an object all the time, e.g., the dog’s eye or nose."

Thank you for pointing out fundamental issues related to the aim of our study. We concluded that the object's center of gravity was appropriate for the following three reasons:

1. The aim of the study was to establish novel metrics to analyze the eye-gaze pattern between the objects and the eye positions.

2. The distance between the eyes and the mouth of the character is so close (the distance was around 15 pixels on 480 x 720 pixels screen) that exact distance between the eccentric part in the characters and eye gaze position was not accurately acquired. 

3. Observation of people with SPD and PIMD revealed that the places where they focus their attention were so diverse that it is difficult to unify them to a specific object.

The present study used the video, in which the characters move well in the field of view, to analyze the eye-gaze position and patterns for the moving objects. To clarify the cognitive functions such as staring at distinct parts of the face, as the reviewer suggested, a future investigation may be necessary to conduct using video images focused on the specific area of the objects (e.g., a face with a close-up of the eyes and mouth) that are large enough to evaluate which feature areas are being viewed by the subjects.

 

Reviewer #2: 

Response: 

We appreciate the time and effort the reviewer has dedicated to providing insightful feedback on ways to strengthen our paper. We have incorporated changes that reflect the detailed suggestions you have graciously provided. We hope that our edits and the responses we provide below satisfactorily address all the concerns the reviewer has noted.

"The research presents a quantified approach for assessment of gaze for people with motor disabilities. The research can be extended to analyze gaze of any individual based on saliency score.

The language of presentation is clear.

The data analysis in the present work show difference of saliency score between healthy individuals and people with MD.

However the technique of video analysis (technique of calculating scoring metrics between video frames) seems cumbersome."

We sincerely appreciate your positive feedback. The saliency map for calculating the saliency score used in this study is based on a standard tool (OpenCV) and can be calculated with simple process. The distance score can also be easily measured from the center of gravity of the moving object and the eye-gaze position. 

We have disclosed all codes for scoring metrics which are available from https://github.com/clinfo/TobiiEyeTrackerVideoAnalysis.git.

---

## [Decision Letter · Decision Letter 1]

1 Dec 2021

PONE-D-21-25819R1Characterizing eye-gaze positions of people with severe motor dysfunction: novel scoring metrics using eye-tracking and video analysisPLOS ONE

Dear Dr. Okuno,

Thank you for submitting your manuscript to PLOS ONE. After careful consideration, we feel that it has merit but does not fully meet PLOS ONE’s publication criteria as it currently stands. Therefore, we invite you to submit a revised version of the manuscript that addresses the points raised during the review process.

We look forward to receiving your revised manuscript.

Kind regards,

Felix Albu, Ph.D.

Academic Editor

PLOS ONE

Additional Editor Comments:

The authors should clearly and thoroughly address the comments of the Reviewer 1.

Reviewers' comments:

Reviewer's Responses to Questions

**Comments to the Author**

1. If the authors have adequately addressed your comments raised in a previous round of review and you feel that this manuscript is now acceptable for publication, you may indicate that here to bypass the “Comments to the Author” section, enter your conflict of interest statement in the “Confidential to Editor” section, and submit your "Accept" recommendation.

Reviewer #1: (No Response)

Reviewer #2: All comments have been addressed

2. Is the manuscript technically sound, and do the data support the conclusions?

Reviewer #1: Yes

Reviewer #2: Yes

3. Has the statistical analysis been performed appropriately and rigorously? 

Reviewer #1: N/A

Reviewer #2: Yes

4. Have the authors made all data underlying the findings in their manuscript fully available?

Reviewer #1: No

Reviewer #2: Yes

5. Is the manuscript presented in an intelligible fashion and written in standard English?

Reviewer #1: Yes

Reviewer #2: Yes

6. Review Comments to the Author

Reviewer #1: Dear Authors, I appreciate your answers. Unfortunately, my concerns were not addressed in the paper, and you took just a minimal effort to improve its quality. I cannot agree that adding three simple sentences is a sufficient improvement after a major review decision. Therefore I assume the paper still needs to be significantly revised.

Reviewer #2: The metrics explained in the manuscript might be helpful for models that are being developed for people with disabilities. The methodology specified might be helpful in gaze analysis. I find the manuscript ready for publication.

7. PLOS authors have the option to publish the peer review history of their article (what does this mean?). If published, this will include your full peer review and any attached files.

Reviewer #1: No

Reviewer #2: No

---

## [Author Response · Author response to Decision Letter 1]

12 Feb 2022

Reviewer #1: 

Response: 

We appreciate the time and effort the reviewer has dedicated to providing insightful feedback on ways to strengthen our paper. We have incorporated changes that reflect the detailed suggestions you have graciously provided. We hope that our edits and the responses we provide below satisfactorily address all the concerns the reviewer has noted.

# Dear Authors, I appreciate your answers. Unfortunately, my concerns were not addressed in the paper, and you took just a minimal effort to improve its quality. I cannot agree that adding three simple sentences is a sufficient improvement after a major review decision. Therefore I assume the paper still needs to be significantly revised.

We apologize for the insufficient description to reflect the insightful feedback of the reviewer. In the previous revision, the reviewer pointed out the following concerns:

1) Details about the system’s accuracy and its impact on the results

2) Information on the tracking system latency, temporal calibration, and offsets.

3) Whether the moving object’s position should be defined as the object’s center of gravity?

We have newly addressed the concerns that the reviewer graciously provided by adding the following paragraph and the table to the Discussion section.

Advantages and limitations of the eye-tracker device in evaluating the gaze responses of people with SPD and PIMD

It is challenging to evaluate eye-gaze responses in people with SPD and PIMD with the same accuracy as healthy subjects. The most confounding factors affecting accuracy are as follows: 1) erratic eye movements are often observed in people with PIMD so that the gaze position shifts with instantaneous amplitude, 2) people with SPD and PIMD are usually hard to follow instructions, 3) people with SPD and PIMD have difficulty maintaining interest in particular objects and need to complete the measurements in a short time. Therefore, to analyze gaze positions in SPD and PIMD patients, eye-tracking devices with high sampling frequency and short, objective observations in an environment suitable for measurement are essential to ensure accuracy. Although eye-tracking could be difficult for conventional devices with low sampling frequency, we used Tobii Pro® Spectrum eye-tracking device to analyze eye-gaze positions in the present study. The tracking system latency of the device was 50 µs that enabled high sampling frequency faster than erratic eye movements (S1 Table).

Temporal calibration and offsets of eye-gaze positions in people with SPD and PIMD are matters, as people with SPD and PIMD have difficulty looking at the instructed points. Therefore, calibration and offset settings were performed by a healthy person instead of each subject. Intergroup comparisons were carried out to reduce measurement error due to the device. In addition, a single observer performed measurements using the same device and video to minimize the measurement time and errors on the observer side. On the other hand, the measurement conditions of the subject differed from person to person because each subject had a different clinical background and spent leisure time differently. So far, no studies have addressed the objective evaluation of the gaze characteristics of people with SPD and PIMD. Although the limitation of measurement errors, the present study has, for the first time, succeeded in obtaining and evaluating the eye-gaze positions and patterns of people with SPD and PIMD, which showed the significance of our novel scoring metrics using eye-tracking and video analysis. The present study was designed to analyze the eye-gaze position and patterns of moving objects that attracted the subject’s interests. The cognitive functions were not addressed, as a distinct position in the face was so close (e.g., the distance between the eye and mouth was about 15 pixels on a 480 x 720-pixel screen) that precise evaluation of what were stared had not been performed precisely. A future investigation may be necessary to conduct using video images focused on the specific area of interest in the objects.

Reviewer #2: 

# The metrics explained in the manuscript might be helpful for models that are being developed for people with disabilities. The methodology specified might be helpful in gaze analysis. I find the manuscript ready for publication.

Response: 

We sincerely appreciate the time and effort the reviewer has dedicated and positive assessment of our research findings.

---

## [Decision Letter · Decision Letter 2]

7 Mar 2022

Characterizing eye-gaze positions of people with severe motor dysfunction: novel scoring metrics using eye-tracking and video analysis

PONE-D-21-25819R2

Dear Dr. Okuno,

We’re pleased to inform you that your manuscript has been judged scientifically suitable for publication and will be formally accepted for publication once it meets all outstanding technical requirements.

Kind regards,

Felix Albu, Ph.D.

Academic Editor

PLOS ONE

Additional Editor Comments (optional):

The paper is accepted.

Reviewers' comments:

Reviewer's Responses to Questions

**Comments to the Author**

1. If the authors have adequately addressed your comments raised in a previous round of review and you feel that this manuscript is now acceptable for publication, you may indicate that here to bypass the “Comments to the Author” section, enter your conflict of interest statement in the “Confidential to Editor” section, and submit your "Accept" recommendation.

Reviewer #1: All comments have been addressed

2. Is the manuscript technically sound, and do the data support the conclusions?

Reviewer #1: Yes

3. Has the statistical analysis been performed appropriately and rigorously? 

Reviewer #1: Yes

4. Have the authors made all data underlying the findings in their manuscript fully available?

Reviewer #1: Yes

5. Is the manuscript presented in an intelligible fashion and written in standard English?

Reviewer #1: Yes

6. Review Comments to the Author

Reviewer #1: Dear Authors, thank you for a more detailed description to my concerns. I think that the new section in the paper improves sound of the paper.

7. PLOS authors have the option to publish the peer review history of their article (what does this mean?). If published, this will include your full peer review and any attached files.

Reviewer #1: No

---

## [Editor Report · Acceptance letter]

9 Aug 2022

PONE-D-21-25819R2 

Characterizing eye-gaze positions of people with severe motor dysfunction: novel scoring metrics using eye-tracking and video analysis 

Dear Dr. Okuno:

I'm pleased to inform you that your manuscript has been deemed suitable for publication in PLOS ONE. Congratulations! Your manuscript is now with our production department. 

Kind regards, 

on behalf of

Dr. Felix Albu 

Academic Editor

PLOS ONE